# Cannabinoid Biosynthesis Using Noncanonical Cannabinoid Synthases

**DOI:** 10.3390/ijms24021259

**Published:** 2023-01-09

**Authors:** Maybelle Kho Go, Tingting Zhu, Kevin Jie Han Lim, Yossa Dwi Hartono, Bo Xue, Hao Fan, Wen Shan Yew

**Affiliations:** 1Synthetic Biology for Clinical and Technological Innovation, National University of Singapore, 14 Medical Drive, Singapore 117599, Singapore; 2Synthetic Biology Translational Research Programme, Yong Loo Lin School of Medicine, National University of Singapore, Singapore 119077, Singapore; 3Department of Biochemistry, Yong Loo Lin School of Medicine, National University of Singapore, 8 Medical Drive, Singapore 117597, Singapore; 4Bioinformatics Institute, A*STAR, 30 Biopolis Street, Matrix #07-01, Singapore 138671, Singapore

**Keywords:** cannabinoid biosynthesis, cannabinoid synthase, berberine bridge enzyme, cannabielsoin, cannabigerolic acid, *Pichia pastoris*

## Abstract

We report enzymes from the berberine bridge enzyme (BBE) superfamily that catalyze the oxidative cyclization of the monoterpene moiety in cannabigerolic acid (CBGA) to form cannabielsoin (CBE). The enzymes are from a variety of organisms and are previously uncharacterized. Out of 232 homologues chosen from the enzyme superfamily, four orthologues were shown to accept CBGA as a substrate and catalyze the biosynthesis of CBE. The four enzymes discovered in this study were recombinantly expressed and purified in *Pichia pastoris*. These enzymes are the first report of heterologous expression of BBEs that did not originate from the *Cannabis* plant that catalyze the production of cannabinoids using CBGA as substrate. This study details a new avenue for discovering and producing natural and unnatural cannabinoids.

## 1. Introduction

*Cannabis sativa* L. is an annual herbaceous plant that has been widely used due to its industrial [1], ornamental [2], and pharmaceutical [3] applications. *Cannabis sativa* contains at least 113 cannabinoid compounds; these cannabinoids exhibit a range of unique pharmacological and therapeutic properties. Despite the report of biosynthesis of tetrahydrocannabinolic acid (THCA) **(2)** and cannabidiolic acid (CBDA) **(3)** using yeast [4], the biosynthetic pathways of most of these cannabinoids remain largely unknown, making it difficult for medicinal cannabinoids to be produced in a stable and sustainable manner [5]. Chemical synthesis of these cannabinoids remains a challenge due to their complex chemical structures. Most of the well-known and studied cannabinoids are derived from cannabigerolic acid (CBGA) **(1)**. Five of the products, THCA **(2)**, CBDA **(3)**, cannabicyclolic acid **(4)**, cannabichromenic acid **(5)**, and cannabicitranic acid **(6)**, are structural isomers (C_22_H_30_O_4_); only cannabielsoic acid B (CBEA) **(7)** is unique, due to an additional oxygen atom (C_22_H_30_O_5_) (Figure 1).

The biosynthesis of cannabinoids from CBGA **(1)** are catalyzed by cannabinoid synthases. The enzymes are characterized as flavin adenine dinucleotide (FAD)—dependent berberine bridge enzymes that catalyze the oxidative cyclization of the monoterpene moiety in CBGA **(1)**. The structure of THCA synthase (Uniprot ID Q8GTB6, PDB ID 3VTE) was elucidated by Kuroki and his associates in 2012 [6]. Cannabidiolic acid synthase was functionally expressed in yeast by Stehle and his associates in 2017 [7]. The known cannabinoid synthases share approximately 80% sequence identity.

Q8GTB6 is an enzyme of the berberine bridge enzyme (BBE) family (IPR012951). The family contains about 4500 members, of which 37 are experimentally annotated and reviewed. Using the EFI-EST tool to generate a sequence similarity network (SSN) of this family [8], it is possible to segregate the proteins into iso-functional clusters. This will place uncharacterized enzymes in sequence–function context with proteins that have been previously characterized from reliable experiments [9,10]. Using this information, it is possible to find potential enzymes from other organisms that will catalyze the oxidative cyclization of the monoterpene moiety in CBGA **(1)**. In this study, we selected 232 homologues from the generated SSN and exploited their potential in accepting CBGA **(1)** as a substrate and catalyzing an FAD-dependent oxidative cyclization reaction to produce novel cannabinoid compounds.

## 2. Results

### 2.1. Sequence Similarity Network of Berberine Bridge Enzymes

The SSN generated approximately 4500 nodes with 300,000 edges (Figure 1). A small number of homologues (232 homologues) were chosen based on either their similarity to Q8GTB6 or their difference. The goal is to create a small yet chemically diverse library to discover novel enzyme activity. The hypothesis of the study follows that those enzymes “similar” to Q8GTB6 may catalyze the production of cannabinoids such as THCA **(2)**; enzymes “different” from Q8GTB6 may catalyze the production of novel cannabinoids. Figure 2 is an illustration of the workflow for the screening of orthologs for cannabinoid synthase activity using CBGA **(1)** as substrate.

### 2.2. Production of Cannabinoids Catalyzed by Potential Cannabinoid Synthase Orthologs

Annotated enzymes from *C. sativa* did not produce detectable amounts of a cannabinoid; it is possible that the signal peptide impeded the heterologous over-expression in *S. cerevisiae* [7]. Most of the homologues chosen (98% of 232 sequences) did not produce any detectable amounts of a cannabinoid, which may indicate that the active sites of BBEs are not as promiscuous as other enzyme superfamilies. Since the organisms chosen for this study are all eukaryotes, it is also possible that the annotated sequences contain introns, which may render the synthesized genes inactive. Three orthologues were discovered to utilize CBGA **(1)** to catalyze the production of CBE **(8)**, the decarboxylated product from CBEA **(7)** (Figure 1). Table 1 shows the Uniprot IDs of orthologues and the organisms they originated from.

M4DIE5 and A0A1J6KPK0 are uncharacterized enzymes. The sequences are annotated as BBE-like enzymes based on sequence homology. P93479 from *Papaver somniferum* was also detected at the transcript level and is proposed to be involved in the formation of (*S*)-scoulerine via the oxidative cyclization of the *N*-methyl moiety of (*S*)-reticuline [12]. The orthologs share about 35–40% sequence identity to Q8GTB6.

The three orthologues were expressed in *P. pastoris* according to previous work on the expression of Q8GTB6 using the same host [13,14]. The purified enzymes were used to determine the kinetic parameters of cannabinoid synthase activity using CBGA **(1)** as substrate (Table 2). When compared to the THCA synthase Q8GTB6 also expressed in *P. pastoris* [13,14], the orthologues are approximately 10-fold slower.

### 2.3. Mass and NMR Analysis of Cannabielsoin

The structure of CBE **(8)** is shown in Figure 3; HR-MS (ESI, negative mode): [M-H]-, *m*/*z* = 329.2134 (experimental); *m*/*z* = 329.2122 (theoretical). ^1^H NMR (CD_3_OD, ppm): H_1_ (1H) 3.3; H_2_ (1H, *m*) 2.5; H_3_ (2H, broad) 1.5; and H_4_ (2H, broad) 1.6; H_6_ (1H, *m*) 3.9; H_7_ (3H, *s*) 1.7; H_9_ (2H, *m*) 4.5 and 4.6; H_10_ (3H, s) 2.0; H_3′_ (1H) 6.1; H_5′_ (1H) 6.1; H_1″_, H_2″_, H_3″_, H_4″_ (8H, *m*) 1.3; H_5″_ (3H, *t*) 0.9, *J*_Ha–Hb_ = 7.5 Hz. ^13^C NMR (CD_3_OD, ppm): C_1_ 49.8; C_2_ 35.7; C_3_ 31.2; C_4_ 31.7; C_5_ 110.3; C_6_ 68.3; C_7_ 17.9; C_8_ 153.5; C_9_ 68.3; C_10_ 26.7; C_1′_ 108.3; C_2′_ 160.3; C_3′_ 143.6; C_4′_ 148.4; C_5′_ 117.1; C_6′_ 163.2; C_1″_ 40.6; C_2″_ 35.7; C_3″_ 34.4; C_4″_ 30.9; C_5″_ 13.0.

## 3. Discussion

Five residues in Q8GTB6 are critical for its activity—two FAD-anchoringresidues, His-114 and Cys-176, which are covalently bonded to FAD, and three catalytic residues, His-292, Tyr-417, and Tyr-484 [6]. Table 3 and Figure 2 show critical residues in Q8GTB6 and the structurally aligned residues in the AlphaFold models of the orthologues. A0A1J6KPK0 contains both the FAD-anchoring residues and the corresponding Tyr-484 residue, whilst His-292 and Tyr-417 are not conserved. The orthologues do not retain amino acid chemical properties for the following positions: His-292 is altered to a hydrophobic residue (Val-291 or Leu-286) and Tyr-417 is changed to an Asn residue (Asn-211, Asn-408, and Asn-394, respectively).

Surprisingly, M4DIE5 lacks both FAD-anchoring residues, only Tyr-484 is conserved. Comparing M4DIE5 AlphaFold model with Q8GTB6 experimental structure (PDB 3VTE), M4DIE5 lacks a whole subdomain that includes the two missing FAD-anchoring residues. M4DIE5 is shorter in sequence compared to the rest and has an RMSD value 1.9 Å over 328 atoms that are structurally aligned to Q8GTB6 (A0A1J6KPK0: 1.7 Å over 496 atoms; P93479: 1.7 Å over 488 atoms). Nevertheless, M4DIE5 has detectable cannabinoid synthase active to catalyze the production of CBE **(7)**.

The three enzymes discovered in this study are the first reported heterologously expressed BBEs that do not originate from the *Cannabis* plant yet can catalyze the production of cannabinoids using CBGA **(1)** as substrate. It is recognized that the study only explored a very limited portion of this enzyme family (232 out of 4500; 5%); the corollary expectation follows that those other enzymes within this family may accept either CBGA **(1)** or/as well as the shorter analogs (with *n*-butyl, *n*-propyl, ethyl, and methyl side chains) as substrates. This study delineates a new avenue for the discovery and biosynthesis of natural and unnatural cannabinoids.

CBEA **(7)** is different from the rest of the carboxylated products in Figure 1 due to an additional oxygen atom; CBE **(8)** is the decarboxylated product of CBEA **(7)** and has previously been reported as an oxidative product from CBDA **(3)** [15]. Analogous to previously determined oxidative cyclization mechanisms of CBGA **(1)**, we propose that it begins with the formation of the carbocation in the monoterpene moiety and the corresponding reduction of FAD to FADH_2_ (Figure 4). The secondary carbocation that is formed rearranges to a more stable tertiary carbocation. Then, a proton is extracted from a terminal methyl group of the octadienyl chain by a general base, and a cascade of electron pair movements forms the cyclohexyl ring. Thereafter, we propose a nucleophilic attack of the first carbocation intermediate (presumably by a nucleophile derived from a H_2_O molecule) to form 2,4-dihydroxy-3-[2-hydroxy-2-methyl-5-(2-propenyl)-cyclohexyl]-6-pentyl-benzoic acid (2,4-OH-6PBz) **(9)**. A second carbocation forms with the corresponding reduction of a second FAD molecule to FADH_2_ and a final cyclization step produces CBEA **(7)**. We propose that a spontaneous decarboxylation occurs to form the final product, CBE **(8)**.

## 4. Materials and Methods

CBGA **(1)** and CBE **(8)** were purchased from Cayman Chemicals. All other chemicals used are of the highest purity that is required for the different experiments. The 232 homologues chosen from the SSN were codon optimized for yeast expression, synthesized, and cloned into pYES2-CT vector (Thermofisher).

The sequence of Q8GTB6 was used as a query to determine the superfamily the enzyme belongs to (http://www.ebi.ac.uk/interpro/ (accessed on 2 January 2018)) [16]. Thereafter, the EFI-EST tool (https://efi.igb.illinois.edu/efi-est/ (accessed on 2 January 2018)) was used to generate the SSN. A small number of homologues (232 homologues) were chosen to test if CBGA **(1)** can be used as a substrate for these enzymes to catalyze the biosynthesis of a cannabinoid. The homologues were chosen using the following criteria: (1) Orthologues found in *C. sativa* and its related organisms such as *Nicotiana* sp.; (2) Orthologues that are experimentally shown to exist either at the protein or transcript level; (3) Orthologues from other plants that share less than 40% sequence identity to Q8GTB6.

The cloned genes were transformed into *Saccharomyces cerevisiae* BY4741 by chemical transformation [17,18,19]. The transformed cells were plated onto SC-URA-glucose plates and incubated for two days at 30 °C. Three single colonies were picked and grown in SC-URA-glucose media for 24 h. The cells were harvested and resuspended in SC-URA-galactose media and incubated for another 24 h. Thereafter, cells were harvested and resuspended in 100 mM citrate, pH 5.5, and 1 mM MgCl_2_. The cell wall was digested by the addition of lyticase (0.5 mg) for 1 h at 37 °C. Glass beads (425–600 μm) were added and the cells were broken using a tissue homogenizer. The cell supernatant was clarified by centrifuge and used for the biosynthetic activity screen.

A 50-μL reaction mixture was prepared for the cannabinoid biosynthesis activity screen. The mixture contains 0.4 mM CBGA, 1.5 mM FAD, and 48 μL cell lysate. The reaction was incubated for 24 h under ambient conditions. The compounds were extracted using ethyl acetate, dried, and re-dissolved in acetonitrile. The presence of the cannabinoids was analyzed using the Agilent RapidFire 365 High-Throughput system coupled with the Agilent 6495 TQ. As mentioned previously, the potential cannabinoid products using CBGA **(1)** have two different molecular formulas—C_22_H_30_O_4_ and C_22_H_30_O_5_ with *m*/*z* values (ESI, negative mode) of 357.2 and 373.2, respectively. Corresponding control experiments in the absence of CBGA **(1)** were also prepared.

Orthologues that were determined to produce the cannabinoid compounds in *S. cerevisiae* were expressed in *P. pastoris* according to published protocols with some modifications [13,14]. A0A1J6KP0 and P93479 were annotated to contain a signaling peptide using the SingalP 5.0 bioinformatic tool. Thus, the signaling peptide was removed and a Strep-tag II and a 6x-histidine tag were added on the C-terminus for purification purposes [20]. The modified genes were cloned into the pPICZα A plasmid (Invitrogen). Cells were transformed by electroporation using 1 mg of *Pme*I linearized plasmid DNA following published protocols (Invitrogen). Transformants were selected by plating the cells on YPD agar plates containing zeocin (100 mg mL^−1^). Clones containing the integrated gene were confirmed by following published protocols (Invitrogen).

The orthologue M4DIE5 does not contain a signaling peptide. Thus, the gene was modified by adding a Strep-tag II on the C-terminus for purification purposes [20]. The gene was cloned into the pPICZα A plasmid (Invitrogen). Cells were transformed and the presence of the gene was confirmed using protocols mentioned previously.

Recombinant *P. pastoris* cells containing the orthologues were grown in BMGY media at 30 °C and 190 rpm for 24 h. Afterwards, cells were harvested by centrifugation at 3000× *g* for 5 min and resuspended in modified BMMY (mBMMY) media to an OD_600_ of 2. Pichia cells were cultivated at 15 °C and 90 rpm and supplemented with 1% (*v*) methanol and 0.02% (*w*/*v*) riboflavin every 24 h for induction of protein expression. Cells were harvested and resuspended in 100 mL of binding buffer. Cells were cracked using a homogenizer (M110P Microfluidizer^®®^) and debris was removed by centrifugation at 20,000 rpm and 4 °C for 30 min. The cleared lysate was loaded onto a Ni^2+^ chelating sepharose column (GE Healthcare) or a Strep-Tactin^®®^ XT column (IBA Lifesciences). For orthologues purified using a Ni^2+^ chelating sepharose column, enzymes were eluted using increasing amounts of imidazole (4 mM to 300 mM imidazole). For orthologues purified using a Strep-Tactin^®®^ XT column, enzymes were eluted using 50 mM D-biotin. Fractions containing the protein were pooled, concentrated, and dialyzed vs. 100 mM citrate, pH 5.5, and 0.2 mM FAD. Purified samples were frozen and stored at −80 °C.

Kinetic parameters of the orthologs were determined using an end-point assay under initial rate conditions. A 100-mL reaction containing 100 citrate, pH 5.5, 0.2 mM FAD, 0.2 to 2 mM CBGA, 5 mM CBS orthologue was prepared for each orthologue. A 20-μL aliquot was stopped by the addition of acetonitrile every 10 min. Samples were filtered and analyzed using the Agilent 1290 Infinity HPLC coupled with the Agilent 6550 iFunnel Q-TOF high resolution mass spectrometer.

The purified proteins were used to catalyze the production of cannabinoids on a larger scale to produce enough compounds for ^1^H-nuclear magnetic resonance (NMR) and ^13^C-NMR analysis. The steps performed are analogous to the smaller scale experiment mentioned previously. Samples were dissolved in CD_3_OD and the spectra were recorded using a Bruker AVANCE 500 MHz NMR spectrometer at the Department of Chemistry, National University of Singapore.

AlphaFold models of the orthologues are taken from their respective UniProt entry pages.

## 5. Conclusions

We have discovered that enzymes from the berberine bridge enzyme family (IPR012951) can catalyze the oxidative cyclization of the monoterpene moiety in CBGA **(1)** to form CBE **(8)**. This is the first report of enzymes that did not originate from the Cannabis plant that catalyze the production of cannabinoids. This study demonstrated the attractive potential of curating SSNs to discover uncharacterized enzymes and curate them based on sequence and structure. Further exploration may include expanding the screen towards the rest of the ~4500 sequences in the BBE family, as well as using the lesser-known analogs of CBGA **(1)** as substrates. We believe that this discovery will aid in the production of cannabinoids in a stable and sustainable manner. This report provides new enzymes that can advance the molecular production of cannabinoid molecules that exist in nature, but for which there are no known molecular tools or enzymes to biosynthesize them. Conceptually, the work also advances the approach to biosynthesize natural products that otherwise are refractory to bioproduction due to the lack of suitably identified enzymes in the biosynthetic pathways.

## 6. Patents

Go, MK; Yew, WS. Biosynthesis of cannabinoids from cannabigerolic acid using novel cannabinoid synthases. WO 2021/071438, 2021.

## Data Availability

The data presented in this study is available within the article.

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
