# Peer review of "Cannabinoid Biosynthesis Using Noncanonical Cannabinoid Synthases"

_ijms, 2023, doi:10.3390/ijms24021259_

Round 1
Reviewer 1 Report
In this manuscript, the authors report berberine-bridge enzyme (BBE) which catalyzes the oxidative cyclization of the monoterpene moiety in cannabigerolic acid (CBGA) and forms the cannabielsoin (CBE). The manuscript seems to be publishable in this journal but some minor concerns must be addressed before its acceptance.
comments
1. NMR/HRMS copies of compounds 7 and 8 should be included
2. Schemes 1 and 3 should be mentioned in the figure
3. In NMR data analyses ppm should not repeat multiple times.
4. typos should be minimized and overall language needs polishing.
5. the stereochemistry of the compounds should be analyzed.
Author Response
We have noted the reviewers’ suggestions for improvement and put forth our responses below. Reviewer comments in Black, our response in Red.
Reviewer #1 (Remarks to the Author):
In this manuscript, the authors report berberine-bridge enzyme (BBE) which catalyzes the oxidative cyclization of the monoterpene moiety in cannabigerolic acid (CBGA) and forms the cannabielsoin (CBE). The manuscript seems to be publishable in this journal but some minor concerns must be addressed before its acceptance.
Comments:
- NMR/HRMS copies of compounds 7 and 8 should be included.
It is not common practice to include NMR spectra of compounds in manuscripts. We have reported the NMR data analyses of compound 8 (CBE), as the manuscript only reports on compound 8. We have uploaded the raw NMR spectra of compound 8 for review only.
- Schemes 1 and 3 should be mentioned in the figure
Schemes 1 and 3 have been mentioned in the manuscript.
- In NMR data analyses ppm should not repeat multiple times.
We have corrected the NMR data analyses.
- typos should be minimized and overall language needs polishing.
We have minimized the typos and polished the overall prose and language of the manuscript.
- the stereochemistry of the compounds should be analyzed.
It is not common practice to analyze the stereochemistry of natural compounds that have been biosynthesized by enzymes. As the manuscript serves to report on the discovery of non-cannonical enzymes to biosynthesize CBE, a compound with no known reported biosynthetic enzyme, we respectfully suggest to the reviewer that our NMR data analyses of compound 8 (CBE) should suffice.
Reviewer 2 Report
In this manuscript, Kho Go et al. reported enzymes from the BBE superfamily that catalyze the oxidative cyclization of the monoterpene moiety in CBGA to form CBE. Although the manuscript is attractive, there are some concerns that should be addressed.
-Generally, the manuscript is well organized but there are some typographical and grammatical errors.
-The paper title is well stated, it is informative and concise.
-Abstract is well structured.
-The introduction was not well written, and it is too briefly presenting the subject and research problem.
L 27: First, introduce cannabis and its application. my suggestion: “Cannabis sativa L. is an annual herbaceous plant that has been widely used due to its industrial (10.3906/bot-1907-15), ornamental (https://doi.org/10.3390/plants11182383), and pharmaceutical (https://doi.org/10.1007/978-981-16-8822-5_4) applications.”
L32: Please provide new reference (s): (https://doi.org/10.1016/j.biotechadv.2022.108074)
L 54-56: the main objectives of the study should be stated.
-Material and research methods are presented appropriately. The experimental setup and the description in the methods section are well structured, and the statistical analysis is correctly performed.
-The results obtained in this study are interesting. Results are presented correctly.
-In general, the discussion was not well written. This part should be improved.
Author Response
We have noted the reviewers’ suggestions for improvement and put forth our responses below. Reviewer comments in Black, our response in Red.
Reviewer #2 (Remarks to the Author):
In this manuscript, Kho Go et al. reported enzymes from the BBE superfamily that catalyze the oxidative cyclization of the monoterpene moiety in CBGA to form CBE. Although the manuscript is attractive, there are some concerns that should be addressed.
-Generally, the manuscript is well organized but there are some typographical and grammatical errors.
We have corrected the typographical and grammatical errors in the manuscript.
-The paper title is well stated, it is informative and concise.
-Abstract is well structured.
-The introduction was not well written, and it is too briefly presenting the subject and research problem.
L 27: First, introduce cannabis and its application. my suggestion: “Cannabis sativa L. is an annual herbaceous plant that has been widely used due to its industrial (10.3906/bot-1907-15), ornamental (https://doi.org/10.3390/plants11182383), and pharmaceutical (https://doi.org/10.1007/978-981-16-8822-5_4) applications.”
We have incorporated the reviewer’s suggestion.
L32: Please provide new reference (s): (https://doi.org/10.1016/j.biotechadv.2022.108074)
We have incorporated the reviewer’s suggestion as reference 5.
L 54-56: the main objectives of the study should be stated.
We have incorporated the reviewer’s suggestion.
-Material and research methods are presented appropriately. The experimental setup and the description in the methods section are well structured, and the statistical analysis is correctly performed.
-The results obtained in this study are interesting. Results are presented correctly.
-In general, the discussion was not well written. This part should be improved.
We have incorporated the reviewer’s suggestion by revising the discussion to improve upon the presentation.
Round 2
Reviewer 1 Report
The authors have improved the manuscript and solved my previous concerns reasonably but NMR data should be clear and we can agree that not common practice in the manuscript. Referees focus always on improving the quality and easy understanding of content by readers. And this revised version can be acceptable. with best wishesReviewer 2 Report
All my comments have been addressed. The current version of the manuscript can be published in IJMS.